# Influence of Plasma-Activated Water on Foliar and Fruit Micronutrient Content and Plant Protection Efficiency

Andrei I. Kuzin [1,2,3,*], Natalia Ya. Kashirskaya [1], Alexei E. Solovchenko [3,4], Anna M. Kochkina [1], Ludmila V. Stepantsowa [2], Vyacheslav N. Krasin [2], Evgeny M. Konchekov [5], Vladimir I. Lukanin [5], Konstantin F. Sergeichev [5], Victoria V. Gudkova [5], Dmitry O. Khort [6] and Igor G. Smirnov [6]

[1] I.V. Michurin Federal Scientific Centre, 393774 Michurinsk, Russia; kashirskaya@fnc-mich.ru (N.Y.K.); ms.anna.step@mail.ru (A.M.K.)

[2] I.V. Michurin Institute of Fundamental and Applied Agrobiotechnologies, Michurinsk State Agrarian University, 393760 Michurinsk, Russia; stepanzowa@mail.ru (L.V.S.); krasin84@yandex.ru (V.N.K.)

[3] Institute of Natural Sciences, Derzhavin Tambov State Agrarian University, 393760 Tambov, Russia; solovchenkoae@my.msu.ru

[4] Faculty of Biology, Lomonosov Moscow State University, 119234 Moscow, Russia

[5] Prokhorov General Physics Institute of the Russian Academy of Sciences, 119991 Moscow, Russia; eukmek@gmail.com (E.M.K.); vladimirlukanin@yandex.ru (V.I.L.); k-sergeichev@yandex.ru (K.F.S.); gudkova-vi@fpl.gpi.ru (V.V.G.)

[6] Federal Scientific Agroengineering Center VIM, 109428 Moscow, Russia; dmitriyhort@mail.ru (D.O.K.); rashn-smirnov@yandex.ru (I.G.S.)

* Correspondence: andrey.kuzin1967@yandex.ru

**Abstract:** Foliar fertilizing is very important to supply apple plants with calcium and micronutrients. The most cost-effective approach to this is the application of the fertilizers in tank mixtures with plant protection chemicals. Plasma-activated water (PAW) has great potential for the use in the agriculture. We used two type of PAWs, PAW1 (made using underwater electrical discharge in an aqueous $KNO_3$ solution and includes reactive nitrogen species and platinum nanoparticles) and PAW2 (made using a plasma torch with nitrogen gas makeup and contains reactive nitrogen species but not metals). We studied the impact of two PAW types on the contents of Ca, B, Mn, Fe, and Co in leaves and Ca, Mn, Fe, Zn, and Mo in fruits sprayed with tank mixtures containing the fertilizers. We also tested the efficiency of PAW in the control of apple scab when applied as tank mixtures with plant protection chemicals. The application of the PAWs significantly increased foliar Ca when the PAW was mixed with Ca-containing formulations (spraying PAW1 containing Ca increased leaf Ca by up to 21%, and PAW2 up by to 9% compared to Ca spraying without PAW). The largest fruit Ca increase was in the variant treated with PAW1 with a micronutrient spraying program (up to 143%). The PAW treatments enhanced the baseline mineral contents of the plants even when they were not sprayed with the nutrients. PAW1 mainly increased the nutrient contents of the apple fruits. PAWs have proven to be efficient for the control of apple scab, thereby reducing the demand for fungicides. The scab damage to the leaves and fruits was similar in plants treated with PAWs without fungicides (1.7–1.9% on the leaves and 1.6–1.8% on the fruits) compared to the conventional chemical scab control (0.9% leaves and 0.6% fruits) and was significantly lower than in the untreated control (9.3% on leaves and 11.9 on fruits).

**Keywords:** plasma-activated water; *Malus × domestica*; *Venturia inaequalis*; calcium; micronutrients; mineral contents in leaves and fruits; scab control; fungicide efficiency

## 1. Introduction

The foliar fertilizing of fruit crops has a long history. Boynton reports on the use of iron sulfate on pineapple already at the beginning of the XX century to overcome iron deficiency resulting from the oversupply of magnesium [1]. The significance of foliar fertilizing with

micronutrients for enhancing plant productivity and stress resistance has been repeatedly emphasized [2–8]. This is especially true for calcium foliar supply for improving the storability of apple fruits [9]. Thus, Hepler [10] and other authoritative references [11,12] argue that the shortage of calcium is deteriorative for apple fruit storability and their consumer quality [13]. An illustrious example is constituted by the role of calcium in the control of bitter pit in apples [14–16].

Calcium is crucial for the prevention of bitter pit in apple. This physiological disorder causes significant economic losses. Bitter pit was shown to be caused by a $Ca^{2+}$ deficiency [17]. The application of Ca-containing foliar fertilizers during the growing season before harvesting can reduce the incidence of bitter pit in apples and improve fruit quality by increasing the Ca content and decreasing the N/Ca ratio in the fruits [18]. This was confirmed by many workers [19–22].

Apple scab, caused by Venturia inaequalis (Cooke) G. Wint., is the most important apple disease, causing economic losses in apple-growing companies [23]. The control of apple scab is permanently in the focus of researchers since the lack of scab damages is a key determinant of fruit commercial value [24]. There are different strategies for the control of apple scab: breeding resistant cultivars [25,26], the use of chemical and biological fungicides [27,28], or combinations thereof. These practices can be combined with chemically compatible foliar fertilizers in tank mixtures to reduce handling and other costs. Many scab control programs aim at protecting trees through the repeated application of fungicides [29,30]. This leads to the development of scab resistance to fungicides [31], and an increase in the pesticide load on the environment [32]. So, the problem of reducing the fungicidal load on the environment while providing reliable apple scab control is very important.

Plasma-activated water is a promising vehicle for making agriculture more environmentally friendly [33]. For PAW preparation, water or water solutions of certain compounds are treated by non-thermal plasma initiated by various discharge types. For example, direct discharge, glow discharge, and plasma jet, among other discharge types, give rise to long-lived reactive oxygen and nitrogen species: hydrogen peroxide, ozone, ions of nitrogen oxides ($NOx^-$), etc. The treatment of plants with PAW allows for the elicitation of effects that are comparable to the direct action of catabolite activator protein (CAP) [34].

PAW enhances plant growth by stimulating seed germination, promoting robust biomass production, and fortifying defense mechanisms against pathogens. Its antimicrobial properties reduce soil-borne pathogens, while improving nutrient uptake and fostering stress resilience. Ongoing research efforts aim to refine PAW application methodologies for optimal agricultural integration.

The application of PAW to a plant can be carried out in various ways. To treat plants against phytopathogens, it is convenient to use sprays (plasma-activated sprays). In this case, a pre-made PAW is sprayed, or the plasma treatment of water is carried out directly at the outlet of the spraying device.

PAW applications stimulated the growth of spruce and strawberry plants [35], as well as black locust tree seeds [36]. PAW also improved the fruit set, yield, leaf nitrogen and potassium contents, fruit phosphorus and calcium contents as well ascorbic acid and titratable acidity of fruits [37]. Dipping tomato in PAW inactivated microorganisms on the fruit's surface [38] and other plant objects [39,40]. Cold plasma treatment did not exert negative effects on Pink Lady® apple fruits [41]. PAW also enhanced pear graft survival [42]. Overall, using cold plasma and its products has the potential for increasing the environmental friendliness of agriculture [43], specifically via improving the effect of fertilizers [44].

We hypothesized that PAW could stimulate an increase in the content of microelements in plant leaves and fruits after tank mix treatments with fertilizers. We also considered the possibility that the combined use of crop protection products and PAWs would not reduce the effectiveness of the phytopathogen control. Since PAWs are applied via spraying, the main objectives of our study were (i) to investigate the impact of PAWs on the effectiveness

of foliar fertilizers with microelements (B, Mn, Fe, Co, Zn, and Mo) and calcium, and (ii) to explore the possibility of using PAWs for the control of apple scab, also in combination with conventional fungicides.

## 2. Materials and Methods

### 2.1. Location and Conditions of the Experiments

#### 2.1.1. Experimental Plots

Experiment #1 was carried out in the experimental orchard of the I.V. Michurin Federal Scientific Centre in Michurinsk (52°53′01.7″ N 40°27′55.2″ E). The objects of this study comprised apple (*Malus* × *domestica* Brokh.) trees of cv. Ligol grafted on B396 rootstock. The orchard was planted in Autumn 2018, with a planting pattern of 4.5 × 1.2 m. Each treatment included 30 trees in three blocks (10 trees per block). The fertigation rate on the experiment plots was $N_{12}P_6K_{20}$ in all treatments during the season 2023 (we reduced the common fertigation rate to highlight the effect of PAW and foliar fertilizing). The application rate is indicated in $kg\,ha^{-1}$ of fertilizer active ingredients. Soil properties are given in Table S1.

Experiment #2 aimed to study the effects of PAW on fungicide efficiency in the control of apple scab; it was also carried out in the experimental orchard of the I.V. Michurin Federal Scientific Centre in Michurinsk (52°53′01.7″ N 40°27′55.2″ E). The objects of the study were apple trees (cv. Spartan) grafted on B118 rootstock. The orchard was planted in Autumn 2015, with a planting pattern of 5.0 × 2.5 m. Each treatment included 20 trees (5 trees per block). Soil management: black fallow, no drip irrigation. The soil fertilizers were not applied in the spring of 2023. Soil properties are given in Table S1.

#### 2.1.2. Weather Conditions

Climatic parameters were recorded by an KaipoRain automatic weather station (Kaipos Ltd., Krasnodar, Russia). The average monthly air temperature in some months was 6.9–21.7 °C (Table 1). The most spectacular deviations from multiyear averages were in precipitation levels. The amount of rainfall was low in April, May August, September, and October, but in June and July, precipitation was normal.

**Table 1.** Average monthly air temperatures (°C) and total monthly precipitation (mm) during the growing season of 2023.

| Month | Temperature, °C | Precipitation, mm |
|---|---|---|
| April | 10.9 | 19.2 |
| May | 12.0 | 15.3 |
| June | 17.7 | 53.2 |
| July | 20.5 | 59.4 |
| August | 21.7 | 6.8 |
| September | 16.3 | 1.2 |
| October | 6.9 | 2.6 |
| Mean IV–X | 15.1 | 22.5 |

### 2.2. PAW Preparation

PAW1 was produced using an electrochemical setup containing an electrolyte vessel fitted with active (platinum (Pt)) and neutral (Pt) electrodes. These electrodes were connected to a high-frequency (HF) generator operating at 440 kHz. The root mean square current was maintained at 0.8 A, with the potential for a peak current of up to 5 A during plasma ignition. The voltage applied to the electrodes was 300 V, and the generator could supply up to 1500 W of power. This power level was sufficient to create a vapor–gas bubble on the active electrode, leading to the initiation of a glow-like discharge in the vapor phase. The electrolyte used in this process was a 2% $KNO_3$ solution, and the experimental reactor had a volume of 6 L. The solution was actively mixed using a magnetic stirrer. The reactor maintained a temperature below 70 °C, which was achieved at an average power input of

240 W during operation. The synthesis of PAW1 took approximately 6 h, during which stable chemical compounds formed in the solution. The active electrode experienced gradual degradation through electrical erosion, resulting in the formation of Pt nanoparticles from the manufacturing material, with sizes ranging from 10 to 20 nm and a concentration of $10^{12}$ particles per milliliter. The presence and dimensions of these nanoparticles were verified using a Zetasizer ULTRA Red Label (Malvern Panalytical, Malvern, UK). The solution can be used as a disinfectant or, when diluted with distilled water (at a ratio of 1:200), for use in plant irrigation [37,44,45]. Table 2 provides additional details about some of the physical and chemical properties of PAW1.

**Table 2.** Physicochemical properties of the PAW.

| Storage Duration, Days | PAW Type | Exposure Time, min | Electrical Conductivity, µS/cm | pH | Redox, mV | $NO_2^-$, µM | $NO_3^-$, mM | $H_2O_2$, µM |
|---|---|---|---|---|---|---|---|---|
| 0 | PAW1 | 360 | 22,806 ± 2281 | 10.8 ± 0.2 | 91 ± 9 | 1447 ± 87 | 215 ± 21 | 0.9 ± 0.5 |
| | PAW2 | 240 | 352 ± 35 | 3.2 ± 0.2 | 500 ± 50 | 209 ± 4 | 3.1 ± 0.3 | 1.8 ± 0.5 |
| 1 | PAW1 | 360 | 22,139 ± 2214 | 10.5 ± 0.2 | 135 ± 14 | 1378 ± 174 | 213 ± 21 | 0 |
| | PAW2 | 240 | 453 ± 45 | 3.2 ± 0.2 | 495 ± 49 | 179 ± 6 | 4.5 ± 0.5 | 0 |
| 2 | PAW1 | 360 | 21,721 ± 2172 | 10.1 ± 0.2 | 171 ± 17 | 1469 ± 46 | 271 ± 3 | 0 |
| | PAW2 | 240 | 456 ± 46 | 3.2 ± 0.2 | 514 ± 51 | 186 ± 15 | 3.9 ± 0.4 | 0 |
| 3 | PAW1 | 360 | 19,809 ± 1981 | 10.0 ± 0.2 | 204 ± 20 | 1414 ± 141 | 287 ± 10 | 0 |
| | PAW2 | 240 | 455 ± 46 | 3.2 ± 0.2 | 504 ± 50 | 116 ± 12 | 4.8 ± 0.3 | 0 |
| 4 | PAW1 | 360 | 21,691 ± 2169 | 9.8 ± 0.2 | 192 ± 19 | 1516 ± 5 | 287 ± 10 | 0 |
| | PAW2 | 240 | 478 ± 48 | 3.2 ± 0.2 | 484 ± 48 | 117 ± 12 | 5.9 ± 0.3 | 0 |
| 7 | PAW1 | 360 | 21,653 ± 2165 | 9.6 ± 0.2 | 211 ± 21 | 1592 ± 7 | 275 ± 17 | 0 |
| | PAW2 | 240 | 475 ± 48 | 3.2 ± 0.2 | 511 ± 51 | 90 ± 21 | 5.0 ± 0.5 | 0 |
| 8 | PAW1 | 360 | 21,697 ± 2169 | 10.4 ± 0.2 | 156 ± 16 | 1530 ± 3 | 247 ± 2 | 0 |
| | PAW2 | 240 | 459 ± 46 | 3.2 ± 0.2 | 508 ± 51 | 64 ± 1 | 4.7 ± 0.3 | 0 |
| 9 | PAW1 | 360 | 22,303 ± 2230 | 10.1 ± 0.2 | 164 ± 16 | 1259 ± 32 | 312 ± 3 | 0 |
| | PAW2 | 240 | 467 ± 47 | 3.2 ± 0.2 | 508 ± 51 | 53 ± 0.3 | 6.1 ± 0.3 | 0 |
| 10 | PAW1 | 360 | 21,845 ± 2185 | 9.9 ± 0.2 | 160 ± 16 | 1369 ± 3 | 255 ± 16 | 0 |
| | PAW2 | 240 | 466 ± 47 | 3.2 ± 0.2 | 502 ± 50 | 42 ± 1 | 5.1 ± 0.2 | 0 |
| 11 | PAW1 | 360 | 21,952 ± 2195 | 9.6 ± 0.2 | 218 ± 22 | 1561 ± 28 | 247 ± 14 | 0 |
| | PAW2 | 240 | 481 ± 48 | 3.2 ± 0.2 | 510 ± 51 | 50 ± 1 | 4.8 ± 0.1 | 0 |

PAW2 was produced using a non-contact method involving an argon plasma jet within a nitrogen atmosphere. The equipment was constructed based on microwave plasma-torch technology. The microwave source utilized was a commercial 1.2 kW power magnetron operating at a frequency of 2.45 GHz in continuous generation mode [46,47]. The plasma jet was created in an isolated chamber with a nitrogen atmosphere using argon as the feed gas. Thus, when the plasma jet interacted with the surface of the distilled water, which was located in a passivated metal tank with a volume of 5 L, reactive nitrogen species were generated in the water. In the process of creating PAW using this method, the emission of metal particles from the electrode is negligible, so this type of PAW can be called "pure", that is, free of metal nanoparticles or compounds. Some physical and chemical properties of PAW2 are presented in Table 2.

In all experiments, the original PAW1 and PAW2 were diluted with distilled water to a concentration of 50 mL $L^{-1}$. We used freshly prepared PAW for spraying.

The concentrations of hydrogen peroxide and nitrite ions were assayed spectrophotometrically using the FOX assay and Griess assay [48–50] by determining the optical density of solutions at a wavelength of 560 nm (reaction time, 5 min) and 525 nm (reaction time, 20 min), respectively. For this purpose, a HACH LANGE DR-5000 spectrophotometer

(HACH LANGE GmbH, Duesseldorf, Germany) was used. $NO_3^-$ ions were detected using a LAQUAtwin NO3-11 (HORIBA Advanced Techno, Kyoto, Japan). The conductivity, pH, and redox potential of the liquids were determined using a SevenExcellence multichannel meter (Mettler Toledo, Greifensee, Switzerland).

*2.3. Experimental Design and Analysis Methods*

Experiment #1 was designed as follows: 1. Control 1 (C1)—without any treatments; 2. Control 2 (C2(ME))—micronutrient foliar fertilizing spraying schedule (Table S2); 3. Control 3 (C3Ca)—calcium foliar fertilizing program (Table S3); 4. PAW1—PAW1 solution; 5. PAW2—PAW2 solution; 6. PAW1(ME)—PAW1 + micronutrient foliar fertilizing program; 7. PAW2(ME)—PAW2 + micronutrient foliar fertilizing program; 8. PAW1(Ca)—PAW1 + calcium foliar fertilizing program; 9. PAW2 + calcium foliar fertilizing program.

We applied the following foliar spraying agrochemicals manufactured by "Schelkovo Agrohim" (Schelkovo, Russia): Ultramag Calcium (Ca 17.0%, N 10.05, MgO 0.8%, Zn 0.02%, Cu 0.02%, B 0.05%, and Mo 0.01%); Ultramag Chelate Fe-13 (Fe-13) (Fe 13.0%); Ultramag Super Zn-700 (Zn-700) (N 1.5% and Zn 40.0); Ultramag Phosohorus Super (N 6.4%, $P_2O_5$ 35.0%, MgO 4.0%, and Zn 2.5%); Ultramag Boron (N 4.7% and B 11.0%); Biostim Growth (aminoacids 4.0%, N 4.0%, $P_2O_5$ 10.0%, MgO 2.0%, $SO_3$ 1.0%, Fe 0.4%, Mn 0.2%, Zn 0.2%, and B 1.0%); Biostim Universal (amino acids 10.0%, N 6.0%, $K_2O$ 1.3%, and $SO_3$ 5.0%); and Ultramag Potassium ($K_2O$ 22.0% and N 2.6%). Information about the active ingredients of these products is presented in Tables S2 and S3. The amount of working solution sprayed once was 333 mL/tree or 617 L/ha.

Leaf and fruit nitrogen (Kjeldahl method, (AKV-20, JSC Villitek, Moskow Russia)), phosphorus (molybdenum blue method (Hitachi U-2000, Hitachi LTD., Tokyo, Japan)), potassium (flame photometer, FPA-2.01, JSC ZOMZ, Zagorsk, Russia), and calcium (complexometric method with trilon B) were assayed. Sulfur was determined using a turbidimetric method (colorimetry with $BaCl_2$ after digesting the sample with acids at a wavelength of 644 nm (Hitachi U-2000, Hitachi Ltd., Tokyo, Japan)). B, Mn, Cu, Zn, Mo, and Co were determined by an atom absorption method (MGA-915MD, Lumex, Saint-Petersburg Russia) [51]. The element content is expressed based on the dry weight.

Each experimental variant consisted of 3 replicas (10 trees each). From each replication (10 trees), 10 fruits (1 per tree) and 40 leaves (4 per tree) from one-year shoots in the middle of the canopy were randomly sampled. Two segments were cut out of each fruit, dried, and ground for the analysis. Four leaves were selected from each tree at the north, south, west, and east sides of canopy, rinsed with distilled water, dried, and ground to powder. Leaves were sampled on 21 August 2023 and fruits on 26 September 2023.

Experiment #2′s design was as follows: Control 1 (C1P)—without protection; Control 2 (C2P)—full protection program (Table S4) without PAW treatments; PAW1 + full protection program (PAW1 + PT); PAW2 + full protection program (PAW2 + PT); PAW1 without protection against scab (PAW1); PAW2 without protection against scab (PAW2).

We applied the following chemicals for foliar sprayings to control phytopathogens: Cuproxat SC (Nufarm GmbH & Co. KG, Linz, Austria); Chorus WG (Sungenta AG, Basel, Switzerland); Fontelis SC (DuPont, Wilmington, NC, USA); Delan WG; Bellis WG (BASF GmbH, Mannheim, Germany); and Medeya ME (Shelkovo Agrohim, Shelkovo, Russia).

The experimental variant consists of 4 replicas (5 trees each). On each tree, 100 leaves and fruits were examined (25 leaves and fruits on 4 branches from two sides). The severity of scab damage to leaves and fruits was assessed visually and assigned points: 0 = no symptoms; 1 = up to 10% of the surface area of the leaves or fruits is damaged, and spots are small and without sporulation; 2 = 11–25% of the surface area of leaves or fruits is damaged; 3 = 26–50% of the surface area of leaves or fruits is damaged; 4 = 51–75% of the surface area of leaves or fruits is damaged [52].

The severity of scab reflects the average degree of damage in the experimental unit (plot) and was calculated using the formula:

$$R = \frac{\sum (a \cdot b) \cdot 100}{N \cdot K} \tag{1}$$

where $R$ is disease severity (%), $\sum(a \cdot b)$ is the sum of the multiplications of the number of diseased plants ($a$) and the corresponding score of damage in points ($b$), $N$ is the total number plants in the sample, and $K$ is the highest score of damages according to the scale.

Abbott's formula was used to calculate the biological efficiency of protection against scab [53]:

$$C = \frac{x - y}{x} * 100, \tag{2}$$

where $C$ is the biological efficiency (percent control);

$x$ is the severity of scab in the untreated control;

$y$ is the severity of scab in the treated plot.

The commercial quality of the fruits was assessed based on the following:

The "Extra Fancy" grade consists of apples of a single variety, typical in color and shape for the given pomological variety, free from damage by pests and diseases, with or without stem, without damage to the skin of the fruit, and free from scab lesions.

The "Fancy" grade consists of apples of one cultivar which are typical or atypical, with less pronounced color, without damage by pests and diseases, with or without a stalk, and without damage to the skin of the fruit. Fruit should have no more than 1–2 scab lesions, each with a diameter of 3 mm or less.

The "Utility" grade consists of apples of one cultivar. Fruits may be heterogeneous in size and color, with or without a stalk. Damage by diseases and pests, including scab (the diameter of lesions is more than 5 mm, and the number is more than 5 per fruit), is allowed.

The main criterion of our assessment of the fruit grade was scab damage severity.

### 2.4. Statistical Treatment

The data were analyzed according to Fisher's method [54]. We calculated the least significant difference (LSD) between the various treatments at $p < 0.05$. The differences that were higher than the computed LSD value were considered to be significant. Statistical data treatment was carried out using Excel 2007 and the AgCXStat add-in [55].

### 3. Results

### 3.1. Leaf and Fruit Calcium

The leaf Ca contents significantly increased upon its application during foliar fertilization (Figure 1a). The highest leaf Ca content was in PAW1(Ca). The increase in leaf Ca status in PAW2(Ca) was also significant, suggesting that the application of this element in the tank mixtures with PAWs boosts the leaf calcium content.

The fruit Ca content in the treatments without additional Ca applied in spraying (C1F, C2F(ME), and PAW2) was at the baseline level [14] (Figure 1b). Expectedly, foliar fertilizing with Ca (C3F(Ca)) significantly increased its content. PAW1 also increased fruit Ca to the level recorded in the C3F(Ca) variant. Interestingly, PAWs caused an increase in the fruit Ca content even in the treatments lacking an exogenic Ca supply, especially PAW1(ME).

### 3.2. Leaf and Fruit Micronutrients

3.2.1. Leaf B, Mn, Fe, and Co Contents

The highest increase in leaf B content was in the treatments solely with PAWs (Figure 2a). Leaf B also increased after supplementation with the micronutrients and Ca (C2F(ME) and C3F(Ca)).

Leaf Mn content varied significantly among the treatments (Figure 2b). Paradoxically, the minimum leaf Mn was noted in the C2F(ME) variant triple-treated with the Mn-containing complex fertilizer. The highest leaf Mn content was observed in the PAW2 and

PAW1 treatments. PAW2(ME) and PAW1(Ca) also exhibited a high content of Mn. At the same time, the Mn content in PAW1(ME) and PAW2(Ca) was significantly lower than that in the C1F variant.

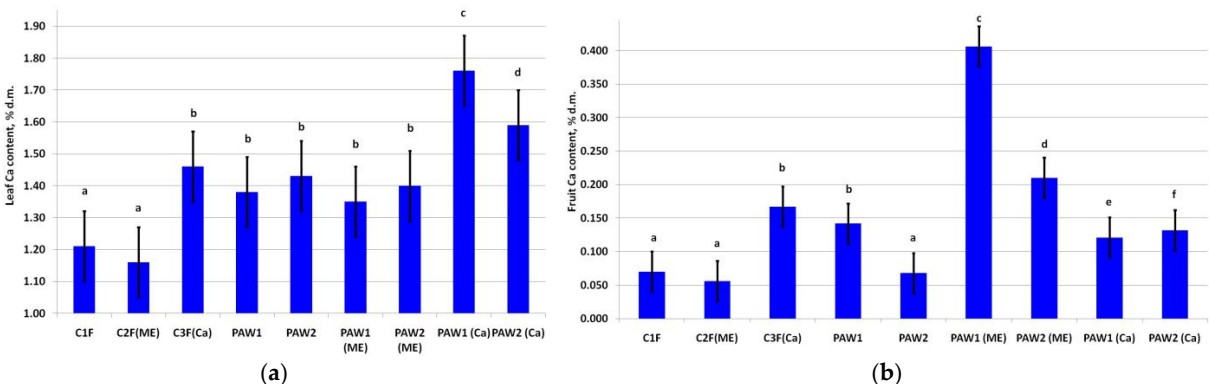

**Figure 1.** Calcium content of the apple (**a**) leaves ($LSD_{05}$ = 0.11% d.m.) and (**b**) fruits ($LSD_{05}$ = 0.03% d.m.). Different letters indicate significantly ($p < 0.05$) different values.

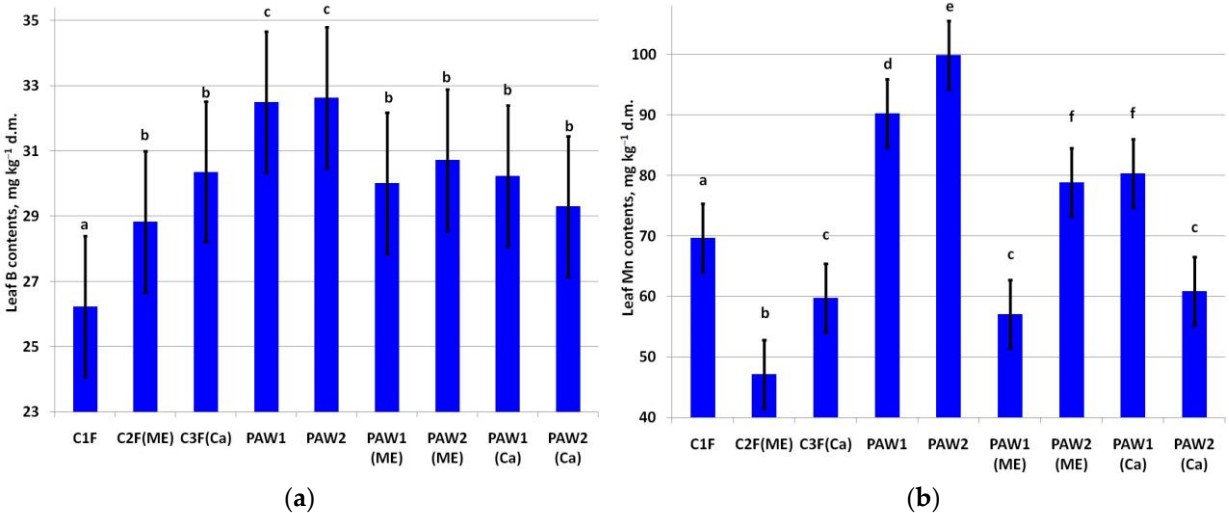

**Figure 2.** Content of (**a**) B ($LSD_{05}$ = 2.16 mg kg$^{-1}$) and (**b**) Mn ($LSD_{05}$ = 5.64 mg kg$^{-1}$) in apple leaves. Different letters indicate significantly ($p < 0.05$) different values.

The leaf Fe content was around the baseline values (50.0–250.0 mg kg$^{-1}$) [56] regardless of the treatment (Figure 3a). A significant increase in leaf Fe was recorded in the C3F(Ca) and PAW2(Me) variants compared to the control and C1F variants. The highest leaf Fe content was in the PAW1(Ca) variant.

A significant increase in leaf Co content was noted in several treatments, but only two variants were prominent in this regard: C3F(Ca) and PAW1(Ca) (Figure 3b).

### 3.2.2. Fruit S, Mn, Fe, Zn, and Mo Contents

The fruit S content increased in C2F(ME) as well as in combination with PAWs with Ca treatments, especially in PAW2 (Figure 4a).

Treatment with exogenic microelements increased fruit Mn content (C2F(ME)), Figure 4b). The largest increase in Mn was noted in PAW2(Ca). The leaf Mn content was also high in the PAW1 and PAW1(ME) variants (Figure 4c). The highest fruit Fe content was found in the PAW1(ME) variant, and in the other treatments, the Fe content varied insignificantly.

Foliar micronutrient fertilizing significantly increased fruit Zn contents in the C2F and PAW1(ME) variants (Figure 5a). The lowest fruit Zn was in the PAW2(ME) variant. The application of the PAWs significantly decreased the fruit Zn content compared to the C1F variant. PAW2 combined with microelements in tank mixtures declined the fruit zinc

content. At the same time, the application of PAW2 in tank mixtures with Ca increased the fruit Zn content significantly compared to both the C1F and C1F(Ca) variants.

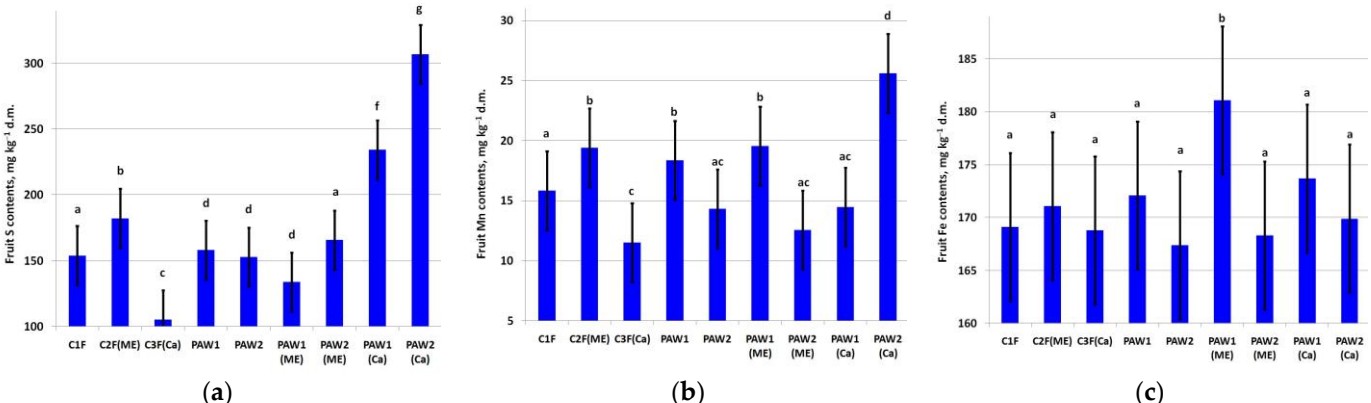

**Figure 3.** Content of (**a**) Fe (LSD$_{05}$ = 9.53 mg kg$^{-1}$) and (**b**) Co (LSD$_{05}$ = 0.004 mg kg$^{-1}$) in apple leaves. Different letters indicate significantly ($p < 0.05$) different values.

**Figure 4.** Content of (**a**) S (LSD$_{05}$ = 22.4 mg kg$^{-1}$), (**b**) Mn (LSD$_{05}$ =3.28 mg kg$^{-1}$) and (**c**) Fe (LSD$_{05}$ = 7.00 mg kg$^{-1}$) in apple fruit. Different letters indicate significantly ($p < 0.05$) different values.

The greatest effect on fruit Mo contents was when PAW1 was applied either with ME fertilizers or Ca fertilizers (Figure 5b).

### 3.3. Influence of PAW on Scab Control Efficiency

The weather conditions during the growing season facilitated scab severity on the leaves more than on the fruits. The level of damage in the controls without treatment was 9.3% (leaves) and 11.9% (fruits, Figure 6a).

The degree of scab severity in the variant with complete chemical plant protection but without PAWs (C2P) was 0.9% on the leaves and 0.6% on the fruits. In the treatments where the fungicides were applied in the tank mixtures with the PAWs, the severity of scab on the leaves was as low as 1.1% (PAW1 + PT) and 1.0% (PAW2 + PT). The degree of severity of the scab on fruits comprised 0.5% (PAW1 + PT) and 0.4% (PAW2 +PT). The application of PAW instead of the fungicides demonstrated interesting results: scab severity on leaves was 1.9% (PAW1 + PTwF) and 1.7% (PAW2 + PTwF). Scab severity on the apple fruits was 1.8% (PAW1PTwF) and 1.6% (PAW2PTwF) in the treatments without fungicides.

Complete protection with chemical fungicides (C2P) provided the highest biological efficiency of scab control on apple leaves (90.0%). The maximum biological efficiency of scab control on fruits was in the treatment PAW2 + PT where plasma-activated water was

used together with the chemical fungicides (96.6%). The application of PAW instead of the fungicides also ensured a higher commercial value of apple than the control without treatments (Table 3).

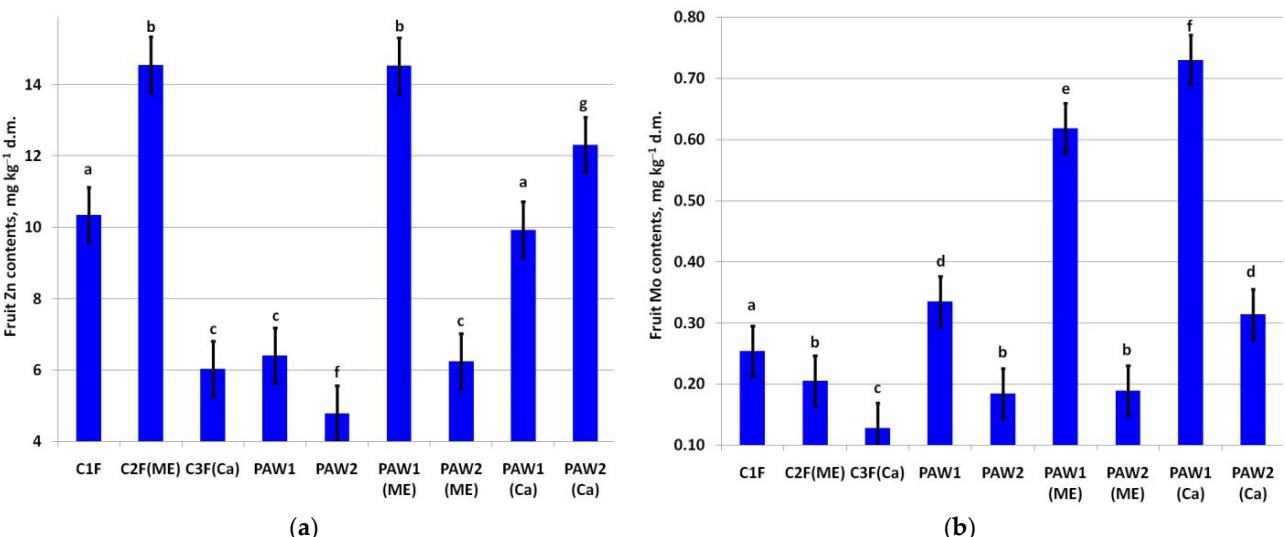

**Figure 5.** Content of (**a**) Zn (LSD$_{05}$ = 0.78 mg kg$^{-1}$) and (**b**) Mo (LSD$_{05}$ = 0.041 mg kg$^{-1}$) in apple fruit. Different letters indicate significantly ($p < 0.05$) different values.

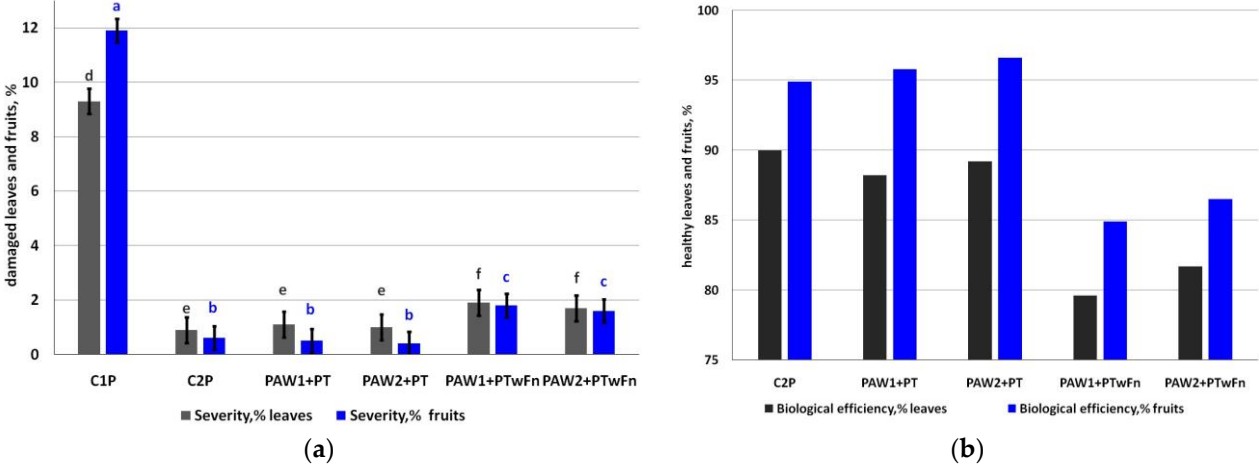

**Figure 6.** Scab severity (**a**) (leaves, LSD$_{05}$ = 0.47, grey letters; fruits, LSD$_{05}$ = 0,43, blue letters) and (**b**) the biological efficiency of apple scab control, %. Different letters indicate significantly ($p < 0.05$) different values.

**Table 3.** Effect of PAWs applied either in tank mixtures with fungicides or individually on the yield and fruit quality.

| Treatments | Yield, kg Tree$^{-1}$ | Extra Fancy, % | Fancy, % | Utility, % |
|---|---|---|---|---|
| C1P | 4.8 [a]* | 10 | 10 | 80 |
| C2P | 6.1 [b] | 88 | 8 | 4 |
| PAW1 + PT | 6.3 [b] | 92 | 5 | 3 |
| PAW1 + PT | 6.4 [b] | 90 | 6 | 4 |
| PAW1 | 6.0 [b] | 60 | 30 | 10 |
| PAW2 | 6.2 [b] | 65 | 25 | 10 |
| LSD$_{05}$ | 0.7 | - | - | - |

* Different letters indicate significantly ($p < 0.05$) different values among the treatments in yield. There are no essential differences among the treatments with the same letter.

## 4. Discussion

The effects of PAW on the mineral contents of plant tissues is scarcely elucidated in the literature [57], as is the interaction of PAW with foliar fertilizers applied in tank mixtures. It is still known that cold plasma has high physiological activity; it can inactivate enzymes such as polyphenol oxidase and peroxidase [58] by the direct oxidation of the enzyme molecules, as was demonstrated by Setsuharo et al., who exposed alanine to argon plasma [59]. Also, cold plasma increased the pH in guava fruits [60], affecting the activity of diverse enzymes [61]. The exact mechanism of the plasma's effects is still unknown [62].

The use of cold plasma for the disinfection of food products revealed a number of problems including the deterioration of quality, the biodegradation of bioactive components, color changes, etc. [62].

In our study, the addition of PAWs to the working solution of the Ca fertilizer increased the nutrient content in leaves and fruits. This was the case even when the PAWs were used without the addition of Ca. These results are in accord with recent studies [37]. Probably, the application of PAW in tank mixtures facilitated the penetration of Ca into the leaves, so the greatest effect was achieved when PAW1 was combined with an elevated concentration of exogenic Ca in tank mixtures. The main source of Ca for plants is its uptake from the soil by roots. Leaves are the main consumer of Ca taken up by the roots [63]. Microelements or surfactants can stimulate the uptake of other nutrients; foliar fertilizing with Mn increased nitrogen uptake, and foliar fertilizing with zinc enhanced K uptake [64]. Presumably, in our experiments PAWs and/or microelements stimulated the uptake of Ca by the roots.

PAW can also increase cytosolic $Ca^{2+}$ levels [65]. Since the application of nitrate, nitrite, and hydrogen peroxide in the concentrations characteristic of the PAW did not trigger detectable $Ca^{2+}$ changes, this phenomenon is likely caused by some other component unique to PAW.

Assorted ions and reactive oxygen species present in PAW can change the charge and structure of the ectodesmata, increasing their permeability. PAW can also influence plants' signaling system, which may result in nutrient redistribution within the plant. It was proved that the PAW irrigation of tomato plants upregulates seedlings growth, endogenous reactive oxygen and nitrogen species, defense hormones (salicylic acid and jasmonic acid), and the expression of key pathogenesis-related (PR) genes [66].

Normally, apple trees are quite responsive to boron foliar fertilizing [67]. In our study, the highest leaf B content was recorded after the treatments with "pure" PAWs, and the addition of PAWs to the working solution also increased leaf B contents compared to the control, C1F. PAW application might improve boron uptake by changing cell signaling.

The leaf Mn content was relatively low in the C2F(ME) variant, likely due to antagonism with other nutrients applied as a complex foliar fertilizer in our spraying program, such as iron [68]. "Pure" PAW application stimulated leaf Mn; PAW1 enhanced the fruit manganese content. Presumably, PAW2 application reduced the antagonistic effect, and in the PAW2(ME) treatment, the leaf Mn content was significantly higher. Interestingly, the highest leaf Mn content was in the PAW1(Ca) treatment, supposedly due to the modulation of root activity. Reports on the mechanism of the PAW modulation of the antagonistic interactions of ions in fertilizers and in plants are scarce, but there is information about stimulating root activity, in particular, increasing the electrical capacity of roots [69].

The leaf Fe content in the C2(ME) treatment was significantly lower than in the control without fertilizer spraying; this could be a result of ionic antagonism as well. The application of PAW2 reduced antagonistic effects, and in the PAW2(ME) treatment, the Fe content was significantly higher. The highest leaf Fe content was in the PAW1(Ca) treatment, presumably due to the effects linked with plant signaling.

Cobalt has low mobility in the leaf tissues [70]; its content in leaves varies from 60 $\mu g\,kg^{-1}$ to 274 $mg\,kg^{-1}$ [71]. The leaf Co content strongly depends not only on the cobalt in the soil but also on the type of soil, which, apparently, can limit its absorption by roots [72]. The leaf cobalt contents in our study was relatively low, but it increased significantly in

the PAW1(Ca) treatment. It could also result from soil Co uptake enhancement due to the impact of PAW1(Ca) on plants' signaling system.

The fruit S content was relatively high in all treatments except for C3F. This could be the result of S/Ca antagonism, according to Reich et al. [73]. PAW application together with Ca fertilizer boosted the sulfur content (especially in the PAW2(Ca) treatment), so the application of PAW seems to reduce the antagonistic S/Ca interaction.

The highest Mn content in the fruits was also observed in the PAW2(Ca) treatment, despite Mn and Ca also being known to be antagonistic [74], albeit not in all cases studied [75]. The former case (Mn/Ca antagonism) was evident in our study, but PAW2 reduced this effect. Regarding Fe content in the fruit, our results were close to those of Škarpa et al.: the effect of PAW on this parameter was insignificant [69].

Overall, PAW can be beneficial for agriculture by promoting plant growth and fortifying plant defense against pathogens (this phenomenon is called "priming") [69,76]. Cold plasma treatments can also modify gene expression. In plasma-treated wheat seeds, the expression of LEMMA1 declined, but that of SnRK2 and P5CS increased [77] (the gene SnRK2 is involved in abscisic acid metabolism [78]). These pieces of evidence showing the effect of PAW on the expression of certain genes may reveal the mechanism of their action to control scab, as well as their influence on plants' signaling system.

The magnitude of PAW's effect on plants is significantly lower than that of cold plasma itself, but the observed PAW effects resemble the modulation of stress-response genes and plant stress signaling between leaves and roots. This response is, in principle, possible under the influence of abscisic acid which is synthesized in leaves and moves through the phloem into the roots [79]. Our findings suggest that the PAW1(Ca) treatment boosted the uptake of all studied microelements, especially iron and cobalt (which were not detected in other experimental variants), likely by modulating the plants' signaling system.

Importantly, using PAW instead of the fungicides was effective in the control of apple scab. The literature contains but scarce information about PAW's impact on apple scab severity, but a well-documented effect of PAW is the inactivation of microorganisms via the induction of oxidative stress [80]. PAW application reduced the spore activity of *Fusarium graminearum*, thereby preventing fungus growth and the infection of wheat seedlings. These effects might be mediated by the PAW-induced cell wall deformation, changes in membrane permeability and mitochondrial dysfunction [81,82].

Admittedly, PAW can inhibit harmful microorganisms through oxidative stress induced by the reactive oxygen species contained therein. Thus, PAW can displace the redox balance and deplete the antioxidants in *Staphylococcus aureus*, eventually killing its cells [83]. Particularly, the reactive oxygen species can destroy important peptidoglycan bonds, which leads to cell wall destruction [84]. Having said this, we do not propose PAW as the sole "replacement" to fungicides.

## 5. Conclusions

PAW considerably increased the leaf Ca contents when it was applied together with a Ca fertilizer. The fruit Ca increased after PAW treatments together with micronutrient fertilizing. Both fruit and leaf micronutrient contents increased after their treatment with PAWs in various combinations with fertilizers. PAW strongly affected the uptake of certain nutrients by plants even without the exogenic application of these nutrients. Thus, PAW1 in combinations with fertilizers mainly affected the nutrient content of fruits. One can assume that the effect of PAW is mediated by alterations of the signaling in plant cells, leading to an increase in the nutrient uptake and redistribution in the plant.

The application of PAW without fungicides was effective in the control of scab and was at the level of conventional chemical fungicides. The use of PAW in tank mixtures with the studied fungicides did not reduce the effectiveness. Accordingly, PAW can be used in plant protection systems to reduce application rates and/or the number of fungicide treatments required to control apple scab.

**Supplementary Materials:** The following supporting information can be downloaded at: https://www.mdpi.com/article/10.3390/horticulturae10010055/s1, Table S1: Soil properties; Table S2: Micronutrient foliar spraying program; Table S3: Calcium foliar spraying program; Table S4: Protection schedule.

**Author Contributions:** Conceptualization, A.I.K., N.Y.K., A.E.S., D.O.K. and E.M.K.; methodology, A.I.K., N.Y.K., L.V.S., E.M.K., K.F.S., V.I.L., V.V.G. and I.G.S.; software, V.N.K., D.O.K. and I.G.S.; validation, A.M.K., L.V.S. and V.N.K.; formal analysis, A.I.K.; investigation, A.I.K., N.Y.K., A.M.K. and L.V.S. resources, E.M.K., N.Y.K. and V.N.K.; data curation, A.E.S.; writing—original draft preparation, A.I.K., A.E.S. and E.M.K.; writing—review and editing, all authors; visualization, A.I.K., A.E.S. and I.G.S.; project administration, A.I.K., D.O.K. and I.G.S.; funding acquisition, A.I.K., D.O.K. and I.G.S. All authors have read and agreed to the published version of the manuscript.

**Funding:** This research was supported by a grant from the Ministry of Science and Higher Education of the Russian Federation for large scientific projects in priority areas of scientific and technological development (grant number 075-15-2020-774), supplementary agreement No. 075-15-2020-774/7 dated 28 June 2023.

**Data Availability Statement:** All the data generated during the study are presented within the article and supplementary materials. The raw data are available from the corresponding author on reasonable request.

**Conflicts of Interest:** The authors declare no conflicts of interest.

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
