# Peer review of "Influence of Plasma-Activated Water on Foliar and Fruit Micronutrient Content and Plant Protection Efficiency"

_horticulturae, doi:10.3390/horticulturae10010055_

Round 1
Reviewer 1 Report
Comments and Suggestions for Authors
Dear Editor, in the manuscript Horticulturae-2773039 the effect of foliar fertilizing with Plasma-Activated Water on foliar and fruit micronutrient content and plant protection efficiency. The manuscript provides new and interesting information and could be suitable for publication, although the following comments should be considered before acceptance:
- Line 23: The two types pf PAW should be addressed.
- Lines- 23-24: B was measured in leaves and not in fruit and Mo was measured in fruit but not in leaves, why?
- Line 52: Some words seem to be missing.
- Line 110: Is Pt correct for active and neutral electrodes?
- Line 152: Where is Table 6? Tables should be numbered in the order that they are described in the text.
- Lines 165-168: Where are all these data show? Please, clarify, because they seem to be not provided.
- Line 178: How many fruit or leaves were taken from each tree?
- Lines 188-189: Please, clarify because the scab damage is expressed as % in Figure 6.
- Line 194-205: Where a<re these data shown? Please, be consistent and describe methods for all the parameters measured and provided in tables or figures and vice-versa.
- Figure 1B. Why Ca content in fruit was higher in PAW1(ME) than in PAW1(Ca)? The contrary would be expected as occurred in leaves.
- Figure 2 b: Why Mn content in fruit was higher for PAW1 and PAW2 treatments (without fertilization) than for PAW1(ME) or PAW2(ME) (in which mineral fertilization was applied?)
- Avoid using personal expressions such as “we think” (line 295) along the manuscript.
- Lines 326-332: Speculative and not supported by literature. Rewrite and cite appropriate references.
- Lines 349-353: No relevant information for this manuscript.
- Lines 356-357: Usually, the contrary occurs. That is to say, ABA is synthetized in roots and transported to leaves through xylem. Please, clarify.
- Lines 374-376: Please, clarify the meaning. Do you mean that PAW decreased the fungicide effects?
- Write references according to the journal format.
Comments on the Quality of English Language
Only minor corrections.
Author Response
Reviewer 1
Dear Editor, in the manuscript Horticulturae-2773039 the effect of foliar fertilizing with Plasma-Activated Water on foliar and fruit micronutrient content and plant protection efficiency. The manuscript provides new and interesting information and could be suitable for publication, although the following comments should be considered before acceptance
RESPONSE: we appreciate the careful analysis and positive evaluation of our manuscript by the reviewer. Please see below our detailed responses to the comments made/questions raised by the reviewer.
- Line 23: The two types pf PAW should be addressed.
RESPONSE: we added the information about different PAW types to the Abstract
- Lines- 23-24: B was measured in leaves and not in fruit and Mo was measured in fruit but not in leaves, why?
RESPONSE: initially, we planned to do all analyzes for all nutrients, both for leaves and fruits. In some cases, very contradictory data were obtained among replicates, so we decided to exclude these data from publication now.
- Line 52: Some words seem to be missing.
RESPONSE: we added the word “tree” in the line 102.
- Line 110: Is Pt correct for active and neutral electrodes?.
RESPONSE: Yes, Pt is correct for active and neutral electrodes.
- Line 152: Where is Table 6? Tables should be numbered in the order that they are described in the text.
RESPONSE: thanks, we corrected this typo.
- Lines 165-168: Where are all these data show? Please, clarify, because they seem to be not provided.
RESPONSE: we deleted the information which is not related to the current study.
- Line 178: How many fruit or leaves were taken from each tree?
RESPONSE: we added the information about the number of leaves and fruits sampled.
- Lines 188-189: Please, clarify because the scab damage is expressed as % in Figure 6.
RESPONSE: First, we assessed scab damage in points, and then calculated R (severity of scab damage) values as percentages according to the formula (1).
- Line 194-205: Where a<re these data shown? Please, be consistent and describe methods for all the parameters measured and provided in tables or figures and vice-versa.
RESPONSE: These data is shown in the Figure 6 a and b. The corresponding methods are described in the Materials and Methods Section, subsection 2.3.
Figure 1B. Why Ca content in fruit was higher in PAW1(ME) than in PAW1(Ca)? The contrary would be expected as occurred in leaves.
RESPONSE: the main source of calcium for plants is absorption from the soil by roots. However, the leaves, which are the attracting organ, are the main consumer of the calcium absorbed by the roots. As shown in other publications, including those from our lab, microelements and other bioactive substances can stimulate the uptake of other nutrients. In our early studies, foliar fertilizing with manganese increased nitrogen uptake, and foliar fertilizing with potassium zinc, presumably, by long-distance signaling (we added this statement to Discussion section).
- Figure 2 b: Why Mn content in fruit was higher for PAW1 and PAW2 treatments (without fertilization) than for PAW1(ME) or PAW2(ME) (in which mineral fertilization was applied?)
RESPONSE: Figure 2b shows the Mn contents in leaves. Several reasons for the low Mn content in variants with fertilizer application: 1) Leaves for analysis were taken more than 1.5 months after the last application of fertilizers, so some aftereffect might be involved; 2) we applied a complex micronutrient fertilizer, which, in addition to Mn, contained Fe, so antagonistic interactions between these nutrients can take plase (it is now in the Discussion); 3) various effects of PAWs have not been sufficiently studied, but they can modify inter alia the Mn uptake. Please see also the revised mansucript with changes highlighted for specific changes made to the text.
- Avoid using personal expressions such as “we think” (line 295) along the manuscript.
RESPONSE: corrected.
- Lines 326-332: Speculative and not supported by literature. Rewrite and cite appropriate references
RESPONSE: we changed this and updated the Discussion correspondingly.
- Lines 349-353: No relevant information for this manuscript.
RESPONSE: In this paragraph, we provided information about the effect of PAWs on the expression of certain genes, which may reveal the mechanism of their action to control scab, as well as influence on the plant signaling system. We consider this information important, although currently available in the literature evidence is indeed too small to consider it proven.
- Lines 356-357: Usually, the contrary occurs. That is to say, ABA is synthetized in roots and transported to leaves through xylem. Please, clarify
RESPONSE: we cited the results of Hu et al.: “Leaf ABA syntheses is triggered at different times during root stress and leaf stress …” illustrating the possible effects of PAW oxidative stress on plant life.
- Lines 374-376: Please, clarify the meaning. Do you mean that PAW decreased the fungicide effects?
RESPONSE: We removed the last sentence to avoid cobfusion.
- Write references according to the journal format.
RESPONSE: we did our best to write references according to the journal format.

Reviewer 2 Report
Comments and Suggestions for Authors
This is an interesting and potentially useful study on the interactive effects of two different types of plasma-activated water (PAW) with different types of liquid fertilizers and fungicides for apple scab control. In my assessment, the manuscript should be acceptable following suitable revisions.
Most importantly, the English writing of the manuscript (grammar, word choice, sentence structure, etc.) needs to be improved. The manuscript is currently not at a stage that would allow publication in an international journal. The authors need to work with a native speaker of English or one of the commercial English language editing services before the paper can be considered further.
There are some statistical idiosyncrasies, particularly regarding the mean separations in Figures 1A, 1B, 2B, 3A, 4A, 4B, 5A, and 5B. Please see specific comments below for details.
Materials & Methods should mention whether the PAWs were made fresh for each application.
The Introduction needs more depth, e.g., on the role of Ca in bitter pit of apple and current challenges of management of apple scab with fungicides. Both topics are directly related to the experiments described in this paper. In addition, the Introduction should explain (briefly) how PAW is generated, what some of the physico-chemical differences are compared with regular water, and potential mechanisms of action when applied to plants. In addition to the objectives of the study, the underlying hypotheses should be mentioned at the end of the Introduction.
Additional comments and suggestions (by line numbers of the PDF):
20-30: In the Abstract, mention how the two types of PAW were generated and/or how they differed from each other. Also, the Abstract should include some numerical data, e.g., the percentage by which foliar Ca was increased or the relative improvement in scab control.
31-32: The authors should give the selection of Keywords a little more thought. For example, “apple leaves”, “apple fruits”, or “calcium” are much too broad. On the other hand, “Malus x domestica” and “Venturia inaequalis” should be added.
71-72: Fertigation implies application of the nutrients via the irrigation system. Is this what was done here? How many kg of N, P, and K were applied per tree or per hectare?
In section 2.2, I assume PAW1 and PAW2 were made fresh for each of the spray applications? This should be stated in the manuscript.
152: Table S3 instead of Table 6.
183-186: Mention the active ingredients in these fungicides.
194 and 196: This is disease incidence, not disease prevalence.
200: This is disease severity, not disease development.
209-211: Mention the software and specific procedure(s) used to analyze the data.
Table 1: In the legend, indicate that the weather data is for 2023.
Figure 1: The mean separation letters seem incorrect. Specifically, in Fig. 1A, c and d seem to be switched (if “a” corresponds to the lowest value, then “d” (not “c”) must be associated with the largest value). In Fig. 1B, c, d, e, and f seem to be switched.
There are also incorrect mean separation letters in Fig. 2B, 3A, 4A, 4B, 5A, and 5B.
In all figure legends, indicate whether the error bars correspond to SD or SE, and the number of data points (replicates) used to calculate means and errors.
Figure 6: This is disease severity, not disease development. In Fig. 6A, add error bars.
A Data Availability Statement needs to be added. How does the interested reader get access to the raw data of this study?
Tables S1 and S5: These tables should go in the main text instead of the Supplement.
Table S1: In a footnote, briefly explain how the two different types of PAW were generated.
Tables S2, S3, and S4: In a footnote, mention the manufacturers or suppliers of the agrichemicals.
Table S5: In a footnote, explain how “Extra fancy” and “Fancy” were determined.
Comments on the Quality of English LanguageAs stated above, the English writing of the manuscript (grammar, word choice, sentence structure, etc.) needs to be improved. The manuscript is currently not at a stage that would allow publication in an international journal. The authors need to work with a native speaker of English or one of the commercial English language editing services before the paper can be considered further.
Author Response
Reviewer 2
This is an interesting and potentially useful study on the interactive effects of two different types of plasma-activated water (PAW) with different types of liquid fertilizers and fungicides for apple scab control. In my assessment, the manuscript should be acceptable following suitable revisions.
RESPONSE: we appreciate the careful analysis and positive evaluation of our manuscript by the reviewer. Please see below our detailed responses to the comments made/questions raised by the reviewer.
Most importantly, the English writing of the manuscript (grammar, word choice, sentence structure, etc.) needs to be improved. The manuscript is currently not at a stage that would allow publication in an international journal. The authors need to work with a native speaker of English or one of the commercial English language editing services before the paper can be considered further.
RESPONSE: we did our best to bring the English of the manusript to the standard of scientific publication.
There are some statistical idiosyncrasies, particularly regarding the mean separations in Figures 1A, 1B, 2B, 3A, 4A, 4B, 5A, and 5B. Please see specific comments below for details.
RESPONSE: pleases see our specific answers below.
Materials & Methods should mention whether the PAWs were made fresh for each application.
RESPONSE: we added the information to the Subsection 2.2., Line 179. Please see the revised mansucript with changes highlighted for specific changes made to the text.
The Introduction needs more depth, e.g., on the role of Ca in bitter pit of apple and current challenges of management of apple scab with fungicides. Both topics are directly related to the experiments described in this paper. In addition, the Introduction should explain (briefly) how PAW is generated, what some of the physico-chemical differences are compared with regular water, and potential mechanisms of action when applied to plants. In addition to the objectives of the study, the underlying hypotheses should be mentioned at the end of the Introduction.
RESPONSE: we agreed with the suggestions of Reviewer 1 and added the necessary information to the Introduction section. Please see the revised mansucript with changes highlighted for specific changes made to the text.
Additional comments and suggestions (by line numbers of the PDF):
20-30: In the Abstract, mention how the two types of PAW were generated and/or how they differed from each other. Also, the Abstract should include some numerical data, e.g., the percentage by which foliar Ca was increased or the relative improvement in scab control.
RESPONSE: we agree with the remark and added necessary information to the Abstract. Please see the revised mansucript with changes highlighted for specific changes made to the text.
31-32: The authors should give the selection of Keywords a little more thought. For example, “apple leaves”, “apple fruits”, or “calcium” are much too broad. On the other hand, “Malus x domestica” and “Venturia inaequalis” should be added.
RESPONSE: we made some changes in our selection of Keywords and excluded “apple leaves”, “apple fruits”, as well as added “Malus x domestica” and “Venturia inaequalis”. We prefer to retain “Calcium” because its application and content in leaf and fruit under PAW treatments is important part of our study.
71-72: Fertigation implies application of the nutrients via the irrigation system. Is this what was done here? How many kg of N, P, and K were applied per tree or per hectare?
RESPONSE: the information about the fertigation is in Materials and Methods section, in the subsection 2.1.1. We added the information according the recommendaton of the Reviewer.
In section 2.2, I assume PAW1 and PAW2 were made fresh for each of the spray applications? This should be stated in the manuscript.
RESPONSE: Yes, we used freshly prepared PAW.
152: Table S3 instead of Table 6.
RESPONSE: thanks, we corrected this typo.
183-186: Mention the active ingredients in these fungicides.
RESPONSE: the information about active ingredients of plant protection products is in the Table S4. (spraying program)”
209-211: Mention the software and specific procedure(s) used to analyze the data.
RESPONSE: we added the the information to the Subsection 2.4.
Table 1: In the legend, indicate that the weather data is for 2023.
RESPONSE: we added the the information to the Table 1 caption.
Figure 1: The mean separation letters seem incorrect. Specifically, in Fig. 1A, c and d seem to be switched (if “a” corresponds to the lowest value, then “d” (not “c”) must be associated with the largest value). In Fig. 1B, c, d, e, and f seem to be switched.
RESPONSE: the “separation letters” in the Figure 1 designate significance of the differences between the corresponding average values. The letters were assigned accordinging to the comparison of actual difference of the average with the pre-calculated LSD (significantly different averages were labelled with different letters and thos differing non-significantly were labelled with the same letters). Please note that these letters have nothing to do with ordering of the labelled values. We added a corresponding explanation to the Figures’ legend.
There are also incorrect mean separation letters in Fig. 2B, 3A, 4A, 4B, 5A, and 5B.
RESPONSE: please see our reply to your previous comment.
In all figure legends, indicate whether the error bars correspond to SD or SE, and the number of data points (replicates) used to calculate means and errors.
RESPONSE: the error bars indicate the LSD value calculated for the entire set of experimental variants (please also see above). For each treatment, we selected samples from each repetition (3 per treatment) for an analysis of the content of elements. Then, analyzes of each sample were also performed in triplicate. Those, number of data points for each variant is 9. We added the LSD vlaue to the Figure legend.
Figure 6: This is disease severity, not disease development. In Fig. 6A, add error bars.
RESPONSE: we rectified the terminology as pointed by the reviewer.
Tables S1 and S5: These tables should go in the main text instead of the Supplement.
RESPONSE: we put Table S1 as Table 2 and Table S5 as Table 3 in the main text of the manuscript.
Table S1: In a footnote, briefly explain how the two different types of PAW were generated.
RESPONSE: this was done in Table 2 (Physicochemical properties of the PAW) in the Subsection 2.2 (PAW preparation).
Tables S2, S3, and S4: In a footnote, mention the manufacturers or suppliers of the agrichemicals.
RESPONSE: the fertilizer manufacturer (both calcium and micronutrients) is listed in Subsection 2.3. “Experimental Design and Analysis Methods” lines 196-197. The manufacturers of the plant protection chemicals are listed in section 2.3. “Experimental Design and Analysis Methods” lines 225-228.
Table S5: In a footnote, explain how “Extra fancy” and “Fancy” were determined.
RESPONSE: we added this information to Subsection 2.2. “Experimental Design and Analysis Methods”, lines 248-258.
As stated above, the English writing of the manuscript (grammar, word choice, sentence structure, etc.) needs to be improved. The manuscript is currently not at a stage that would allow publication in an international journal. The authors need to work with a native speaker of English or one of the commercial English language editing services before the paper can be considered further.
RESPONSE: we did our best to bring the English of the manusript to the standard of scientific publication.

Reviewer 3 Report
Comments and Suggestions for Authors
The concept of using PAW to increase uptake of micronutrients is interesting and data suggest that this was successful, at least for leaves. There are several areas that need to be addressed before this paper is useful for readers. First are details on what PAW is, what apple scab is and potential limits of fungicides, why spraying was used when some references refer to dipping (goes back to explaining what PAW is). There is a lot of detail on the soil types-this can be moved to a table unless this has some sort of effect on micronutrient uptake/scab development. No abbreviations are provided-I am assuming Pt refers to platinum? And again, could use a paragraph or 2 explaining PAW and why 2 ways of making PAW were employed.
When were sprays applied? Only before bloom? After bloom?
Apple scab severity and incidence was clearly low. If values were normalized, I suspect that no treatments would differ significantly. At most, you might have a small trend but it would need at least another year or 2 to get a strong picture.
There is value in the micronutrient data but discussion darts around from point to point and no story emerges.
And finally, avoid red and green, suggest using blue, black, gray bars. For color blind people, red and green do not resolve well.
Comments on the Quality of English LanguageThe English overall is good but I keep picking up gaps where a thought is started but then another thought is added in without really finishing and connecting these thoughts. This I think is more of an English issue than a science or writing issue.
Author Response
Reviewer 3
The concept of using PAW to increase uptake of micronutrients is interesting and data suggest that this was successful, at least for leaves. There are several areas that need to be addressed before this paper is useful for readers.
RESPONSE: we appreciate the careful analysis and positive evaluation of our manuscript by the reviewer. Please see below our detailed responses to the comments made/questions raised by the reviewer
First are details on what PAW is, what apple scab is and potential limits of fungicides, why spraying was used when some references refer to dipping There is a lot of detail on the soil types-this can be moved to a table unless this has some sort of effect on micronutrient uptake/scab development.
RESPONSE: the preparation and composition of the PAW is described in the section 2.2. “PAW preparation” and in the Introduction. Spraying allows to treat the whole plants in the field which is not possible with dipping. We agree with the suggestion of the respected Reviewer 2 and table S1 with soil properties has been moved to the Supplementary. Please see also the revised mansucript with changes highlighted for specific changes made to the text.
No abbreviations are provided-I am assuming Pt refers to platinum? And again, could use a paragraph or 2 explaining PAW and why 2 ways of making PAW were employed.
RESPONSE: We added the explanation that Pt refers to platinum.
When were sprays applied? Only before bloom? After bloom?
RESPONSE: the spraying schedule is presented in Tables S2-S4. The left column if stage of plant growth and th last column is the date when they were applied in the season 2023. PAWs were appled in all these sprayings, except of control treatments according to experiment designs (C1F, C2F(ME), C3F(Ca), C1P, C2P). Sprayings with mineral nutrients started on 20.04.2023 (18 days before bloom, first spraying) before bloom and finished on 06.07.2023 (47 days after bloom, eighth spraying). Sprayings with calcium started on 10.05.2023 (blooming stage, first spraying) before bloom and finished 15.09.2023 (117 days after bloom, twelfth spraying). Sprayings with plant protection chemicals started on 11.04.2023 (27 days before bloom, first spraying) before bloom and finished on 09.08.2023 (80 days after bloom, nineth spraying).
Apple scab severity and incidence was clearly low. If values were normalized, I suspect that no treatments would differ significantly. At most, you might have a small trend but it would need at least another year or 2 to get a strong picture.
RESPONSE: We agree with the remark of respected reviewer 2, based on this study, we can only discuss the establishment of a certain trend associated with the use of PAWs in mineral nutrition and protection of apple trees. The use of surfactants in agriculture is a fairly new and interesting topic, and there is not enough information in the literature on numerous issues. Therefore, we consider it necessary to publish the results of our research to fill the information gaps.
There is value in the micronutrient data but discussion darts around from point to point and no story emerges.
RESPONSE: С точки зрения Вашего покорного слуги спорный вопрос.
And finally, avoid red and green, suggest using blue, black, gray bars. For color blind people, red and green do not resolve well.
RESPONSE: done.
Comments on the Quality of English Language
The English overall is good but I keep picking up gaps where a thought is started but then another thought is added in without really finishing and connecting these thoughts. This I think is more of an English issue than a science or writing issue.
RESPONSE:
We would like to sincerely thank all the reviewers; these recommendations have been taken into account in the revised version of the manuscript.

Round 2
Reviewer 1 Report
Comments and Suggestions for Authors
The manuscript has been revised and amended according to the reviewers' suggestions and it could be suitable for publication in its present from.